# The Role of Orthogonality in Genetic Code Expansion

**DOI:** 10.3390/life9030058

**Published:** 2019-07-05

**Authors:** Pol Arranz-Gibert, Jaymin R. Patel, Farren J. Isaacs

**Affiliations:** 1Department of Molecular, Cellular, and Developmental Biology, Yale University, New Haven, CT 06520, USA; 2Systems Biology Institute, Yale University, West Haven, CT 06516, USA

**Keywords:** genetic code expansion, translation, nonstandard amino acids, genome recoding, ribosome engineering, orthogonality, protein engineering

## Abstract

The genetic code defines how information in the genome is translated into protein. Aside from a handful of isolated exceptions, this code is universal. Researchers have developed techniques to artificially expand the genetic code, repurposing codons and translational machinery to incorporate nonstandard amino acids (nsAAs) into proteins. A key challenge for robust genetic code expansion is orthogonality; the engineered machinery used to introduce nsAAs into proteins must co-exist with native translation and gene expression without cross-reactivity or pleiotropy. The issue of orthogonality manifests at several levels, including those of codons, ribosomes, aminoacyl-tRNA synthetases, tRNAs, and elongation factors. In this concept paper, we describe advances in genome recoding, translational engineering and associated challenges rooted in establishing orthogonality needed to expand the genetic code.

## 1. Introduction

Species across all three domains of life share a “universal” genetic code. This code, comprising the full set of 64 tri-nucleotide codons, determines how messenger RNAs are translated into polypeptide chains composed of the 20 standard amino acids (sAAs). Despite its high degree of conservation, the genetic code exhibits flexibility. Natural deviations in the code have evolved, most notably in mitochondrial genomes, where the function of specific codons is often reassigned [1,2]. In some instances, certain codons lack assignment altogether, such as in the bacterium *Micrococcus luteus,* which entirely lacks NNA (N = A, C, G, or T) codons and the tRNAs to decode them [3]. In other cases, a codon can be reassigned to an entirely new amino acid; many species can encode the 21st amino acid selenocysteine [4], while archaeal *Methanosarcina* and bacterial *Desulfitobacterium* strains can encode the 22nd amino acid pyrrolysine [5]. There is also flexibility in triplet decoding, with numerous genes encoding frameshifts within the reading frame [6].

Drawing inspiration from nature’s flexibility, researchers have sought to synthetically expand the genetic code to include new amino acid chemistries into proteins by engineering translational machinery and the genetic code itself [7]. In this field, an underlying theme has been orthogonality, the challenge of expanding the genetic code to introduce nonstandard amino acids (nsAAs) into proteins without perturbing or cross-reacting with endogenous gene expression, or the native translational apparatus. One strategy for nsAA incorporation is to use native translational machinery to incorporate amino acid analogs; for instance, azido-homo-alanine can be incorporated at AUG codons upon methionine depletion in culture [8]. A major orthogonality constraint here is that the nsAA substitutes a natural amino acid throughout the proteome. To achieve more robust orthogonality, strategies have focused on engineering aminoacyl-tRNA synthetase (aaRS)/tRNA pairs, whole-genome recoding to open up dedicated coding channels, and creating isolated ribosomal pools for the synthesis of abiotic peptide chemistries. We describe strategies and challenges associated with achieving orthogonality in the field of genetic code expansion.

## 2. Orthogonal Translation Systems

The major thrust of research in genetic code expansion has focused on the development of “orthogonal translation systems” (OTSs), which consist of (1) an engineered aminoacyl tRNA synthetase (aaRS) that charges (2) a nonstandard amino acid (nsAA) onto (3) its cognate tRNA [9]. The charged tRNA can then promote incorporation of the nsAA at the ribosome into proteins.

Orthogonality is an important consideration when developing new OTSs (Figure 1). First, the aaRS must be selective for its cognate tRNA and vice versa. To achieve this, OTSs are typically derived from aaRS/tRNA pairs sourced from phylogenetically distant organisms. This can help minimize cross-reactivity between the newly-introduced and native aaRS/tRNA pairs, since phylogenetic distance can lead to divergence in the tRNA identity elements that are recognized by the aaRS [10]. Many of the OTSs developed for bacterial expression are derived from archaeal and eukaryotic sources. For instance, the commonly-used TyrRS-tRNA^Tyr^ pair from the archaea *Methanococcus jannaschii* uses the C1-G72 base-pair on the tRNA as a major identity element, whereas *E. coli*’s tRNA^Tyr^ has a G1-C72 base-pair [11]. Thus, the archaeal and eubacterial Tyr systems are mutually orthogonal in this case. Similarly, the yeast *S. cerevisiae* tRNA^Trp^ contains a sufficiently divergent accepter stem that is orthogonal to the Trp system in *E. coli* [12]. Although phylogenetic distance provides a sufficient first-pass level of orthogonality, many OTSs have been further rationally engineered to reduce cross-reactivity [13].

Second, the amino acid binding pocket of the aaRS is diversified to achieve orthogonality at the aminoacylation level. Positive and negative selections are performed to select for aaRS variants that selectively charge the nsAA over the standard amino acids (sAAs), thus achieving orthogonality from endogenous amino acids.

The complexity of OTS engineering is compounded when attempting to simultaneously use multiple OTS systems within the same cell. Recently, three distinct nsAAs have been simultaneously incorporated into a single protein [14]. To achieve this, the multiple aaRS/tRNA pairs need to be orthogonal to native translational machinery as well as to each other. This can be done by using OTSs sourced from very divergent origins, minimizing cross-reactivity. Progress has also been made in rationally engineering mutual orthogonality through acceptor stem changes [15]. An added challenge is that many designed OTSs possess polyspecificity among nsAAs [16]. For instance, TyrRS-based OTSs often can charge numerous diverse tyrosine analogs. Along these lines, recent work has also demonstrated the ability to limit the nsAA specificities of aaRS enzymes through directed evolution to permit the simultaneous use of multiple nsAAs in a single strain [17].

## 3. Orthogonal Genetic Codes

Efficient and multi-site incorporation of nsAAs into proteins require dedicated coding channels. It has long been observed that the anticodons of tRNAs could be mutated to incorporate amino acids at stop codons, effectively turning a stop codon into a sense codon, a process termed suppression [18]. In genetic code expansion efforts, the tRNA of an OTS is similarly engineered to suppress the amber stop codon (UAG). The nsAA is incorporated at in-frame amber codons within an mRNA transcript. However, this approach leads to two key challenges of orthogonality. First, this approach leads to global suppression of all amber codons in the transcriptome, not just in the gene-of-interest. This results in cytotoxicity and decreases the overall efficiency of the OTS. Second, in the case of amber suppression, the OTS competes with release factors at the amber codon; this results in ambiguous decoding with truncated peptides as the dominant product.

Bolstered by recent advances in genome engineering technologies—both in multiplex site-directed mutagenesis and through de novo DNA synthesis of whole genomes [19,20,21,22,23]—substantial work has focused on whole-genome recoding to construct orthogonal genetic codes. Here, the degeneracy of the genetic code is exploited by reassigning all instances of a codon to a synonymous codon, thus freeing up a coding channel for nsAAs while preserving the natural function of that codon [24]. In 2013, the first genomically recoded organism (GRO) was constructed by reassigning all 321 UAG stop codons in *E. coli* to the synonymous UAA [25]. Since this was a synonymous codon swap, the native proteomic sequence was unperturbed. Subsequently, the *prfA* gene, encoding the release factor 1, was deleted, removing any native ability to decode the amber codon and terminate protein synthesis. Thus, the amber codon was converted from a stop to a sense codon, establishing an orthogonal, fully dedicated coding channel for the site-specific incorporation of nsAAs in protein [25]. Although earlier studies demonstrated that prfA can be deleted by minimal recoding of the UAG-terminating essential genes [26,27] or engineering prfB [28], these strategies result in cytotoxicity and are hampered by required complementation of an amber suppressor or peptidyl cleavage by RF-2 at the UAG codon, respectively. Collectively, these studies establish the advantages for engineering an orthogonal dedicated coding channel for efficient and multi-site incorporation of nsAAs in proteins.

Further recoding attempts, through the reassignment of sense codons, have opened up additional coding channels [29,30,31,32] (Figure 2). With sense codon reassignment, it is important to note that due to wobble base pairing during translation, many codons are recognized in pairs by the same tRNA and must be recoded together to achieve orthogonality. It can also be challenging to develop orthogonal tRNAs for sense codon reassignment because aaRSs often use the anticodon as a major identity element. For instance, a synthetic suppressor tRNA with a CCG anticodon can be mischarged by the arginyl-aaRS, which natively recognizes CCG anticodon loops [33]. By decoupling the engineering of OTS from cell viability in vitro translation systems afford greater flexibility in genetic code expansion [34]. For example, the replacement of specific endogenous tRNAs by OTSs or tRNAs pre-charged with a nsAA enabled the incorporation of nsAAs using sense and stop codons [35,36].

In complementary efforts, researchers have sought to use four- and five-base codons to substantially increase the number of possible coding channels for nsAA incorporation [37]. However, decoding mRNA with expanded codons can be inefficient due to orthogonality constraints. Decoding in quadruplet and quintuplet codons competes with decoding in the standard triplet frame. In addition, the presence of quadruplet suppressor tRNAs can cause transcriptome-wide frameshifting events, leading to off-target mistranslation and toxicity. In this regard, suppressing the quadruplet codon AGGA proved moderately successful. Presumably because the AGG codon is quite rare in *E. coli*, the AGGA suppressor tRNA faced lesser competition and had fewer off-target effects [38]. The efficiency of such quadruplet decoding can be further enhanced through ribosomal mutations, as described below. The GRO, lacking all TAG codons, also enhanced quadruplet decoding. In this strain, TAGN codons are suppressed since the chief competition, release factor 1, has been deleted [39].

Drastic expansions in the number of coding channels can be achieved by introducing entirely new nonstandard nucleobases (nsNBs) to the genetic code. To this end, the Hirao group developed a hydrophobic nsNB that is replicated in vitro [40,41]. The Romesberg group has also developed a nonstandard base-pair based on hydrophobic interactions that was mutually orthogonal to the standard four bases [42]. These new bases can be replicated, transcribed, and decoded with engineered OTSs in vivo [43]. In vivo incorporation of these nsNBs posed various challenges. A heterologous nucleobase transporter was isolated and introduced to allow the bases to enter *E. coli*. To augment long-term stability of these nsNBs in vivo, CRISPR/Cas9 was employed to kill cells that had lost the nsNB pair [44]. The nature of the interaction of these nsNBs forces them to adopt a noncanonical pairing, the hydrophobic pairing produces a partial inter-strand intercalation that distorts the structure of the DNA [45,46]. These structural differences with the four standard bases, which adopt the Watson–Crick pairing, have probably limited their use to the middle position of the codon. In contrast, Benner and co-workers designed orthogonal nsNB pairs that rely on hydrogen bonding for base-pairing. These synthetic nucleobases, termed Z, P, S, and B, are thermodynamically stable and do not appear to alter the structure of the DNA double helix. These bases are successfully transcribed in vitro with a mutant T7 RNA polymerase [47]. Expanding this work in vivo has the potential to significantly expand the number of codons, but may pose additional challenges, such as the import of nonstandard nucleobases and replication by native DNA polymerase.

## 4. Orthogonal Ribosomes

Native ribosomes have demonstrated substantial flexibility in tolerating diverse nsAAs [48,49]. In addition to nsAAs, which contain novel side-chain chemistries, there is interest in incorporating entirely distinct, non-L-α-amino acid monomer units into proteins. Such diverse monomers may be excluded by the ribosome during translation. Mutations can be made to the ribosome to allow it to tolerate these exotic monomers; however, these mutations can drastically perturb native translation and impair cellular fitness. To overcome this issue, most of the recent efforts are based on in vitro translation [50,51]. Thus, to enable an in vivo system for polymerizing monomers with abiotic chemistries, it becomes essential to create orthogonal ribosomes, thereby separating the engineered ribosomal pool from native translation.

Early efforts have developed ‘orthogonal’ ribosomes with reduced affinity to native mRNAs [52]. Two challenges have been faced when pursuing this goal: (1) establishing orthogonality between native ribosomes and engineered mRNAs, and between engineered ribosomes and native mRNA; and (2) minimizing cross-assembly between native and engineered subunits of the ribosome. The first challenge has been addressed by modifying the 3′ terminal anti-Shine Dalgarno sequence of the small ribosomal subunit as well as the Shine Dalgarno sequence on the orthogonal mRNAs, thus altering the specificity of translation initiation. This enables the creation of orthogonal ribosome-mRNA pairs [53] whereby mutations have been introduced in the small subunit of this orthogonal ribosome that favor certain suppression events, most notably the suppression of UAG and AGGA codons as channels for nsAA incorporation [54]. These mutations, if made to native ribosomes, would have been toxic to the cell. Therefore, by engineering orthogonality, the effects of these mutations are directed to the “orthogonal mRNA”.

The second challenge has been addressed by tethering the large subunit to the orthogonal small subunit through linker sequences [55,56]. The physical merging of the small and large subunits enables the creation and evaluation of otherwise-lethal mutations in the large subunit. For instance, these mutations can alter the peptidyl transferase center (PTC) of the ribosome to permit new bond formation beyond amide bonds or expand the diversity of permissible amino acid chemistry. Current efforts are applying this technology to enable incorporation of novel backbone chemistries into proteins, such as D-amino acids and other nonstandard monomers that deviate from the native L-α-amino acids.

There remain several challenges associated with orthogonal ribosomes. Current designs have shown reduced activity and incomplete orthogonality [56,57]. The relatively weak activity, potentially due to changes in the structure and folding dynamics, must be compensated by overexpression. This, in turn, imposes a fitness burden on cells, which creates a selective pressure against this orthogonal machinery. Additionally, orthogonality is not strictly observed—native ribosomes translate orthogonal mRNAs and vice-versa; and orthogonal subunits have been shown to interact with native counterparts.

As these limitations begin to be addressed, orthogonal ribosomes will likely play a key role in template-directed production of polymers with expanded chemistries, thus pushing the boundaries of the translation apparatus and expanding the genetic code to polymerize new classes of materials (Figure 3).

## 5. Conclusions

Genetic code expansion has enabled the incorporation of more than 150 nsAAs into proteins. In these efforts, orthogonality between the native and engineered translation machinery is a key parameter that enables robust and high-fidelity incorporation of nsAAs. This issue is compounded when attempting to incorporate multiple nsAAs into multiple positions within a single protein or polymer. At the codon level, whole-genome recoding and the introduction of fully synthetic nucleobases has created dedicated, orthogonal coding channels for nsAAs. Rational engineering of aaRSs, tRNAs and other translation factors have created OTSs with greater orthogonality; however cross-reactivity is often still observed, leading to issues with translational fidelity. The diversity of genetic code expansion will be further enhanced by optimized orthogonal ribosomes, which can segregate nsAA incorporation into a dedicated translational pool, leaving native translation unperturbed. These technologies synergistically combined will enable the next-generation platform for genetic code expansion. We envision that these recent advances in genome recoding, translation machinery and ribosome engineering will lead to a new paradigm for biomanufacturing new sequence-defined biopolymers and biomaterials comprising diverse chemistries that span natural and abiotic chemistries.

## Figures and Tables

**Figure 1 life-09-00058-f001:**
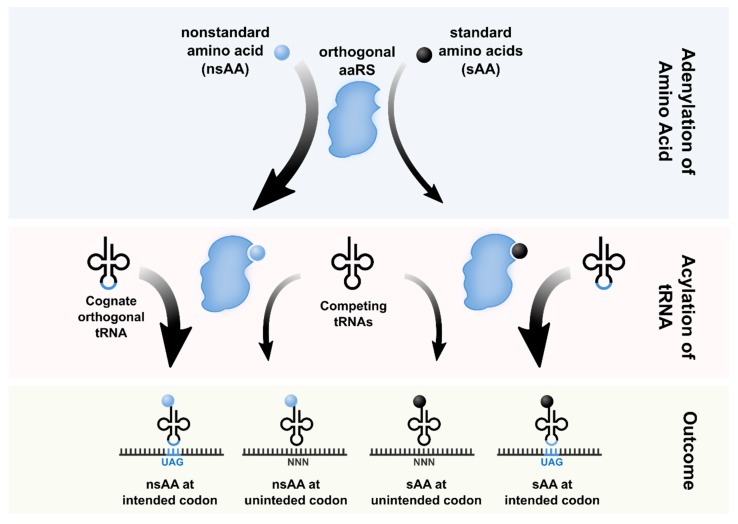
Orthogonality involved in the aminoacylation of tRNAs. First, (blue) an orthogonal aminoacyl-tRNA synthetase (aaRS) can aminoacylate the nonstandard amino acid (nsAA) or a chemically close standard amino acid (sAA); then, (orange) each aaRS∙AA pair can aminoacylate its orthogonal tRNA or another one of the native pool; (green) all of these possible AA-tRNAs leave four potential translation outcomes.

**Figure 2 life-09-00058-f002:**
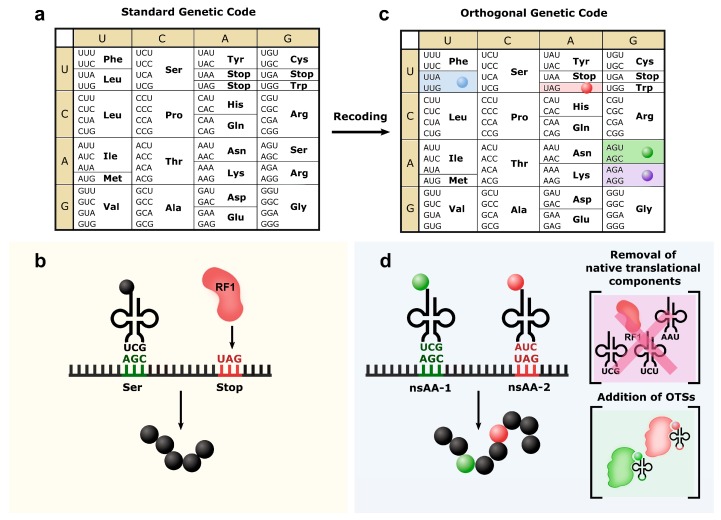
Recoding repurposes the usage of codons. (**a**) The “universal” genetic code contains 61 sense codons and three stop codons. (**b**) (green) AGC and (red) UAG are used as sense codon for serine incorporation and stop codon, respectively. (**c**) These codons can be reassigned to other codons with the same meaning: (blue) leucine, (green) serine, (purple) arginine and (red) UAG stop codons have been reassigned to synonym codons. (**d**) By removing the native components that use the specific codons and introducing an orthogonal translation system (OTS), a new nsAA can be introduced into proteins using the new open codons: while (green) AGC and (red) UAG formerly encoded for serine or stopping translation, respectively, they are now used to introduce two different nsAAs.

**Figure 3 life-09-00058-f003:**
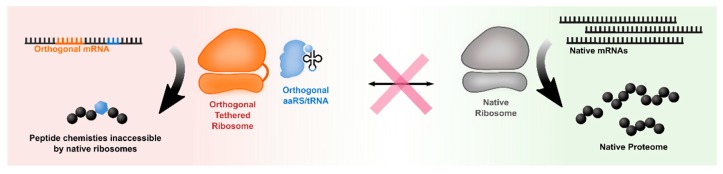
Using orthogonal ribosomes to segregate nsAA incorporation into select proteins. Ideally, when native and orthogonal ribosomes coexist in a cell, they do not interact. The native transcriptome is translated by wildtype ribosomes, whereas orthogonal mRNAs are selectively translated by orthogonal tethered ribosomes (oRiboT). These mRNAs contain reassigned codons that are suppressed by an nsAA-charged tRNA.

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
