# Peer review of "The Role of Orthogonality in Genetic Code Expansion"

_life, 2019, doi:10.3390/life9030058_

Round 1

Reviewer 1 Report

This is a fine review over the recent advances in the field of genetic code expansion.

I suggest a revision concerning the following point.

The authors mentioned an achievement in removing RF-1 from recoded E. coli cells for UAG codon reassignment (l. 97). RF-1 KO does not necessarily require a full recoding of the genome to eliminate UAG codons entirely, as reported in the following two papers, which convey an interesting insight into the nature and flexibility of the genetic code. These papers should be referred to in the revised manuscript.

1.   Mukai, T. et al. Codon reassignment in the Escherichia coli genetic code. Nucleic Acids Res. 38, 8188–8195 (2010).

2.   Johnson, D. B. et al. RF1 knockout allows ribosomal incorporation of unnatural amino acids at multiple sites. Nat. Chem. Biol. 7, 779–786 (2011).

Reviewer 2 Report

Review of:

[Life] Manuscript ID: life-542561

The Role of Orthogonality in Genetic Code Expansion

by Pol Arranz-Gibert, Jaymin Patel and Farren J. Isaacs.

This well-written, rather interesting, very concise, concept paper deals with orthogonality in genetic code expansion. It describes theoretical concepts and the relevant experimental work to specifically change RNA triplet code-amino acid (AA) correlations and even work to create non-triplet RNA or nonstandard nucleobases-AA combinations. Such novel combinations with nonstandard amino acids can be obtained in living cells (or in in vitro extracts): essentially reprogramming the genetic code. Because many levels have to be changed in consort, while perturbation of normal function has to be minimized (!), this constitutes a tall order.

            Overall, the article succeeds in presenting a lot of rather complicated subject matter in a compact, clear and accessible fashion. To begin with a (very) positive aspect: I really like the highly informative figures with all the basic/essential information. But (most of) the legends could be a bit improved; e.g. in figure 2 explain all (!) color codes.

However, though I am a fan of concise papers, the authors could expand their explanations a bit in some locations (specific instances mentioned below). This leads to my only real (still relatively minor) problem with the paper: the combination of the amount of quite different approaches to obtain bio-orthogonality which have to be/are described and its highly concise nature, which makes the text too elliptic at times. Related to this: (i) a few precise definitions of the most important concepts and (ii) not using certain concepts interchangeably might further improve the manuscript (examples given below).

For a review that tries to mention many different approaches in the field I oddly missed the method described in: https://pubs.acs.org/doi/10.1021/acssynbio.6b00245.

Here researchers developed Escherichia coli in vitro translation systems depleted of specific endogenous tRNAs. The translational activity of such a system becomes dependent on addition of synthetic tRNAs (e.g. loaded with nsAA) for the chosen sense codon.

Mostly minor (language) concerns/errors:

Abstract:

Line 10/11: “Aside from a handful of isolated exceptions, this code is largely universal.” “largely” is completely redundant. So: “Aside from a handful of isolated exceptions, this code is universal.”

Line 16/17: “The issue of orthogonality manifests at several levels, including at the level of codons, ribosomes, aminoacyl-tRNA synthetases, tRNAs, and elongation factors.” Should read: “The issue of orthogonality manifests at several levels, including those of codons, ribosomes, aminoacyl-tRNA synthetases, tRNAs, and elongation factors.”

Introduction, etc:

~ Line 40: Bio-orthogonality is also used to describe the incorporation of nsAA’s (such as azido-homo-alanine instead of methionine at AUG codons upon M depletion in culture to allow the specific isolation of newly synthesized polypeptides; see e.g.:

https://www.ncbi.nlm.nih.gov/pmc/articles/PMC2761887/). It might be worthwhile to mention that here and use this to help sharpen the definition of orthogonality you use in the article at this moment.

Line 74/75: “Progress has also been made to rationally engineer mutual orthogonality through acceptor stem engineering [14].” Should read: “Progress has also been made in rationally engineering mutual orthogonality through acceptor stem changes [14].”

Line 88/89: “Second and in the case of amber suppression, the OTS competes…” Should read: “Second (in the case of amber suppression), the OTS competes…”

Line 91: A very brief description of the techniques used in references [18-22] might be appropriate.

Line 106/107: “In such cases, sense codon suppressor tRNAs may cross-react with the corresponding native aaRS resulting in the incorporation of an undesired AA [29].” This sentence is really too elliptic (I had to check reference 29 to be sure I had got the meaning right). Please rewrite in a much more clear fashion.

Line 144: Why the sudden use of  “monomers” instead of AAs? This is just confusing. If the authors want to stress that “strange” AA(like?) molecules can be used, just state so: e.g. “nsAA’s”

Line 160: “These mutations, if made to native ribosomes would have…” Should read: “These mutations, if made to native ribosomes, would have…”

Line 164 (legend figure 3): “Native translation occurs like in the wt cell, whereas…” Should read: “Native translation occurs as in wt cells, whereas…”

Line 173” nonstandard monomers (nsMs)” Why the introduction of an acronym that you do not use in the rest of the tekst?
